# Parental Mental Health, Gender, and Lifestyle Effects on Post-Pandemic Child and Adolescent Psychosocial Problems: A Cross-Sectional Survey in Northern Italy

**DOI:** 10.3390/ijerph21070933

**Published:** 2024-07-17

**Authors:** Verena Barbieri, Giuliano Piccoliori, Adolf Engl, Christian J. Wiedermann

**Affiliations:** 1Institute of General Practice and Public Health, Claudiana College of Health Professions, 39100 Bolzano, Italy; 2Department of Public Health, Medical Decision Making and Health Technology Assessment, University of Health Sciences, Medical Informatics and Technology, 6060 Hall, Austria

**Keywords:** COVID-19, mental health, children and adolescents, screen time, SDQ, anxiety, depression, parental influence, gender differences, sports activities

## Abstract

Background: The exacerbation of psychosocial problems among children and adolescents during the coronavirus disease 2019 (COVID-19) pandemic necessitates an assessment of the long-term mental health impact of pandemic interventions. Focusing on both pandemic-related factors and demographic variables, such as gender and daily habits, an analysis was conducted to understand how these elements continue to affect young populations in the post-pandemic era. Methods: In April 2023, a comprehensive online survey was administered to families in South Tyrol, Italy, with children aged 7–19 years, to ensure age and gender representation. The survey included parent ratings and adolescent (11–19 years) self-reports using standardized instruments to measure the symptoms of mental health problems (Strengths and Difficulties Questionnaire, SDQ), anxiety (Screen for Child Anxiety Related Emotional Disorders, SCARED), and depression (Patient Health Questionnaire-2, PHQ-2). Statistical analyses included descriptive statistics, chi-square tests, and unadjusted odds ratios (ORs). Results: Of the 4525 valid responses, 1831 were self-reported by adolescents. Notable gender differences in mental health outcomes were identified, along with significant demographic predictors, such as age, single parenthood, parental mental health problems, and immigrant background. Negative effects were associated with reduced family climate and increased screen time, whereas physical activity showed beneficial effects. Proxy reports overestimated adolescents’ mental health problems, whereas self-reports tended to underestimate them. Conclusions: Persistent mental health problems and gender disparities highlight the need for a public health approach. This should include accessible support services, resilience building, targeted support for vulnerable families and gender-specific interventions.

## 1. Introduction

The mental well-being of children and adolescents has been closely investigated, particularly as a result of the profound disruptions caused by the COVID-19 pandemic [1]. The impact of such global crises extends far beyond physical health, including the psychological, emotional, and social dimensions of well-being. Worldwide, the lasting effects of the pandemic have been observed, illustrating a diverse spectrum of mental disorders. These range from increased anxiety and depression to changes in social interactions and overall mental health [2,3].

Despite a certain diversity of findings, there is a clear consensus regarding the negative impact of the pandemic on young people’s mental health. Research conducted across Europe provides a picture of mental health outcomes among adolescents in the post-pandemic period, revealing gender differences and an increase in neuropsychiatric symptoms [4]. The dynamic and evolving nature of these challenges emphasizes the need for continued research into the psychological impact of the pandemic, particularly as societies begin to recover and cope with subsequent global crises [5].

In Germany, a comprehensive nationwide longitudinal study covering three years after the onset of the pandemic showed that the prevalence of mental health problems and anxiety escalated from 2020 to 2022 compared with the pre-pandemic period [6]. Italy showed similar trends. Research has pointed to parental anxiety during the pandemic as a contributing factor to the decline in children’s mental health [7,8], accompanied by an increase in hospital admissions for neuropsychological disorders [9]. Nevertheless, more in-depth research is needed in Italy to monitor these health parameters after the pandemic, especially in culturally diverse regions, such as South Tyrol bordering Austria and Switzerland. This analysis is essential for identifying both similarities and differences in the findings compared with those documented in other regions.

During the pandemic, two cross-sectional studies conducted in South Tyrol highlighted that the symptoms of anxiety and depression among local children and adolescents remained consistently high and significantly exceeded pre-pandemic levels [10,11]. The observed trends of increased mental health problems were similarly confirmed in other European countries, where the rates of such problems were significantly lower before the pandemic [12,13]. These studies primarily used a questionnaire developed and initially tested in Germany in the COPSY study [14].

The theoretical framework of this study emphasizes the significant psychological, emotional, and social impacts of global crises such as COVID-19 that have exacerbated mental health problems among adolescents. The present study examined the mental health of children and adolescents in South Tyrol after the pandemic. Building on previous research that identified demographic and pandemic-related factors influencing mental health symptoms [10,11], the 2023 survey focused on (i) testing the hypothesis that, despite a return to “normal” life, the lingering psychological and emotional effects of the pandemic continue to affect young people’s mental health; (ii) confirming the influence of sociodemographic factors, with a particular emphasis on gender differences in mental health outcomes; (iii) analyzing discrepancies between parental (proxy) reports and self-reports of adolescent mental health, including cases in which parents report their own mental health problems and self-reports of adolescents are lacking; and (iv) adolescents’ leisure activities with the aim of identifying simple lifestyle changes that can promote improvements in mental health.

## 2. Materials and Methods

### 2.1. Study Design and Sample

This study used a cross-sectional design to assess mental health among children and adolescents in South Tyrol in April 2023 using the SoSci Survey Software, Version 3.2.46 (SoSci Survey GmbH, Munich, Germany). Following the approach of German COPSY studies [14], the methodology aimed to provide a comparable approach to reflect the evolving nature of the impact of the pandemic on youth mental health. For consistency, the survey was designed to capture the current impact of the pandemic using survey items similar to previous iterations [10,11], but with an emphasis on the unique circumstances of that year. The goal was to provide an up-to-date snapshot of mental health trends among the region’s youth and contribute to a broader understanding of the middle- and long-term impacts of the pandemic.

The participants were students aged 7–19 years from public schools across the province, and informed consent was obtained from their parents or guardians. For children aged 7–10 years, parent questionnaires were completed by parents or guardians. Adolescents aged 11–19 years had the option of completing self-report forms after their parents had completed the proxy version. The parents of the students were contacted by school directorates by email and invited to answer the questionnaire.

The sampling method accurately reflected the age and gender demographics of the South Tyrolean student population according to regional statistics [15]. The methodology remained consistent with previous surveys, but included minor enhancements, such as improved invitation reminders. The response rate for the survey was 23%, approximately 75% of which could be analyzed after data cleaning.

### 2.2. Measures

#### 2.2.1. Sociodemographic Variables, COVID-19 Burden, Family Climate, Digital Media Use, Sports Activity, and Parental Mental Health

Sociodemographic variables included age and gender of children and parents, information about urban/rural residency, single parenthood, migration background, parental mental health, and parental educational attainment, based on the Comparative Analysis of Social Mobility in Industrial Nations (CASMIN) index [16,17].

Parents were asked about their own perceived pandemic-related burden (items on a 5-point Likert scale ranging from 1, “not at all”, to 5, “completely”). Both parents and adolescents were asked about the children’s perceived pandemic-related burden (5-point Likert scale from 1, “not at all”, to 5, “completely”), perceived family climate compared to before the pandemic (5 point-Likert scale from 1, “much worse”, to 5, “much better”), and the actual number of days with more than 1 h of sports activity per week (from 1, “0 days”, to 8, “7 days”).

The actual use of digital media was reported by parents and adolescents as the total number of hours per day spent on computers, smartphones, tablets, and gaming consoles. This included separate evaluations for media use related to school tasks and for private purposes (7-point Likert scale 1, “not at all”, and 2, “less than 1 h”, to 7, “5 h or more”). Additionally, parents and adolescents were asked to compare current media use with their usage before the COVID-19 pandemic, indicating whether it was “much less,” “somewhat less,” “about the same,” “somewhat more,” or “much more.”

Parental mental health was assessed using a parent-reported measure that included two components. Firstly, parents evaluated their own mental health status by responding to a series of questions about how frequently they experienced various symptoms over the past week. These symptoms included lack of interest or pleasure in activities, feeling down or hopeless, sleep disturbances, fatigue, changes in appetite, low self-esteem, concentration difficulties, and changes in motor activity. The response scale ranged from “not at all” to “nearly every day.” Secondly, parents were asked if they had any currently diagnosed mental health conditions by a healthcare professional.

#### 2.2.2. Mental Health

The following instruments were used to assess different aspects of mental health in children and adolescents.

Screen of Child Anxiety Related Emotional Disorders (SCARED): The Generalized Anxiety Disorder (GAD-9) subscale, a tool to measures symptoms of generalized anxiety, was used in the children’s version, asking 9 questions like “I worry about other people liking me.” It uses a 3-point response scale (0, “not or hardly true”, to 2, “very or often true”) and has been widely tested and used [18,19,20]. A total score of nine or more indicated symptoms of generalized anxiety.

Patient Health Questionnaire-2 (PHQ-2): This questionnaire asks 2 questions for depressive symptoms, “Over the last 2 weeks, how often have you been bothered by the following problems: Little interest or pleasure in doing things?” and “Feeling down, depressed, and hopeless?” on a 4-point Likert scale (from 0, “nearly never”, to 3, “nearly every day”) [21,22]. A total score of three or more indicated symptoms of depression.

Strengths and Difficulties Questionnaire (SDQ): The SDQ [23,24] assesses the mental well-being of children and adolescents across five dimensions, each consisting of five questions: emotional symptoms, conduct problems, hyperactivity/inattention, peer relationship problems, and prosocial behavior. The parental version of the questionnaire was used in this study. The total problem score and subscale sum scores were categorized according to official cut-offs into three groups: noticeable/abnormal, borderline, and normal. Higher scores indicate more symptoms of mental health problems. In the results section, the categorization was dichotomized, summarizing the noticeable/abnormal and borderline cases.

### 2.3. Data Analysis

Descriptive statistics were presented as means (M) and standard deviations (SD), while categorical variables were presented as absolute counts and percentages. Differences between groups were assessed using chi-square tests and odds ratios (OR) with 95% confidence intervals (CI) for dichotomous variables. Metric and ordinal variables were compared using the Mann–Whitney U test, and correlations were determined using Spearman’s correlation coefficient. Comparisons between paired samples of proxy and self-reported data were performed using the Wilcoxon rank-sum test.

The main outcomes included symptoms of depression, anxiety, emotional and conduct problems, hyperactivity, peer problems, prosocial behavior, and an overall elevated SDQ score. These were all analyzed as dichotomous variables, with 1 indicating the presence of symptoms and 0 indicating their absence.

Significance levels were set at 0.05, 0.01, and 0.001, and the Bonferroni correction was applied for multiple testing. All statistical analyses were performed using the IBM SPSS Statistics for Windows (version 25.0; IBM Corp., Armonk, NY, USA).

## 3. Results

A total of 6044 parents participated in the survey, and 4525 (74.9%) questionnaires were deemed valid for analysis. Among these participants, 3037 parents reported having children aged 11–19 years. The number of self-reports collected from adolescents in this age group was 1828, constituting 60.2% of the 3037 participants.

### 3.1. Sample Characteristics

Table 1 shows the baseline characteristics of the study participants. Factors such as the age and gender of the children, migration background, and single parenthood are representative of the broader South Tyrolian population. In the survey, age, gender, and residency were mandatory. For all the other parameters in the proxy reports, the missing data rates ranged from 4% to 13%. In the self-reports, the rate of missing answers varied between 6% and 15%.

### 3.2. Mental Health and Pandemic-Related Outcomes by Age and Gender

Table 2 provides a gendered analysis of mental health outcomes among children and adolescents by comparing non-demographic predictor variables between genders, as demographic variables did not show significant gender differences. Overall, a significantly higher percentage of girls reported elevated symptoms of depression and anxiety, with slightly higher levels of depression and almost twice the percentage of anxiety symptoms than boys.

In children aged 7–10 years, parents reported significantly higher levels of conduct problems and hyperactivity in boys, whereas other symptoms showed no gender differences. Among adolescents (11–19 years), proxy reports indicated significantly more emotional problems in girls, while boys were more likely to report symptoms of hyperactivity, peer problems, and difficulties with prosocial behavior. Overall, parents reported symptoms of hyperactivity in both age groups and problems with prosocial behavior in adolescents at a nearly 3:2 ratio for boys to girls.

The analysis delineated dichotomized non-demographic predictors by gender to assess the impact of the pandemic on the lifestyles of children and adolescents. In both age groups, and for both proxy and self-report measures, no significant gender differences were observed for pandemic-related items. Gender differences emerged in lifestyle behaviors. Gender differences emerged in self-reported screen time use of more than three hours per day for school-related concerns among adolescents, with girls reporting higher rates than boys. A similar pattern was observed in proxy reports for the 11–19 years age group, with girls reporting more screen time for school-related concerns than boys. Both self-reports and proxy reports revealed significant gender differences in physical activity levels, with boys consistently engaging in more sports than girls in both age groups.

### 3.3. Unadjusted Odds Ratios for Mental Health Outcomes with Potential Predictors

Unadjusted odds ratios (ORs) were first calculated for dichotomized mental health outcome variables (1 = symptomatic, 0 = not symptomatic), adjusting for potential demographic, pandemic-related, and non-pandemic-related predictors. Variables, both demographic and non-demographic, that had ORs significantly different from 1 are detailed in Table 3.

The results indicated that the mental health of children and adolescents was significantly influenced by all pandemic-related factors. The perceived burden of the pandemic and pandemic-related deterioration in the family climate had the highest odds ratios for symptomatic mental health outcomes in both proxy and self-report measures. Specifically, the highest OR for pandemic-related distress was 9.24, associated with depressive symptoms, followed by an OR of 6.36 for symptoms of emotional problems, and an OR of 5.87 for an overall symptomatic SDQ. Regarding the effect of lower family climate, the highest OR observed was 6.05 for the total symptomatic SDQ, followed by ORs of 5.23 for symptoms of emotional problems and 4.01 for symptoms of conduct problems.

The analysis showed that increased use of digital media for private purposes was associated with an increased OR for all mental health outcomes, whereas use for educational purposes was negatively associated with symptoms of conduct problems and hyperactivity. Increased involvement in sports activities was significantly associated with fewer symptoms in all mental health categories, except for hyperactivity, for which no significant effect was observed.

Among the demographic factors, only a few showed significant ORs for mental health outcomes. Age showed a negative association with symptoms of conduct problems and hyperactivity and a positive association with symptoms of anxiety, depression, emotional problems, and prosocial problems. Although significance levels of 0.05 and 0.01 were frequently achieved, the level of 0.001 was rarely reached. Notably, when Bonferroni-adjusted for multiple testing, factors such as urban residence, low parental education level, and immigrant background were not significant.

Without adjustment for multiple testing, urban residence was only positively associated with symptoms of anxiety, and low parental education was associated with symptoms of emotional problems and overall symptomatic SDQ. Single parenthood was significantly associated with symptoms of depression, emotional problems, peer problems, problematic prosocial behavior, and the total symptomatic SDQ. Similarly, immigrant background was significantly associated with symptoms of depression, emotional problems, hyperactivity, peer problems, and total symptomatic SDQ.

A prominent sociodemographic factor was parental mental health problems, which were significantly associated with all parent-reported mental health outcomes but not self-reported outcomes. Symptoms of conduct problems were significantly associated only with age and gender, whereas symptoms of emotional problems were associated with all demographic factors, except urban residence.

### 3.4. Adolescent Self-Reports and Their Association with Parent Reports for Predictors and Outcomes

The availability of self-reports from adolescents aged 11–19 years was 60.2% relative to proxy reports. The analysis focused on determining whether this percentage was significantly associated with parent-reported predictors and outcome variables. Specifically, we examined whether there were differences in predictors and outcomes between groups of proxy reports that had corresponding adolescent self-reports and those that did not.

#### 3.4.1. Variability in Self-Report Availability and Its Association with Mental Health Outcomes in Adolescents

A significant decrease in the availability of self-report questionnaires was observed with increasing age, lower family climate related to the pandemic, and lower frequency of engaging in more than 60 min of physical activity per day (Figure 1). For the parent-reported outcome variables (SDQ total and subcategories), significant associations were found between the availability of self-reports and fewer symptoms of emotional problems, conduct problems, increased prosocial behavior, and a non-symptomatic total SDQ score. No association was found between hyperactivity and peer problems.

No significant correlations were found between the availability of self-reports and other demographic factors such as the gender of children and parents, parental age, single parenthood, migration status, urban/rural residence, parental education, or parental mental health problems. The pandemic-related factors of adolescent burden and other variables, such as parental pandemic burden and increased digital media use compared to pre-pandemic levels, and the number of hours spent on screen time for private and school purposes not related to the pandemic were not significantly associated with the availability of self-reports.

Within the adolescent age group of 11–19 years, the analysis revealed that the OR for elevated symptoms of emotional problems in the absence of self-report was 1.43 [1.19;1.72] (*p* < 0.001). For symptoms of conduct problems, the OR was 1.24 [1.03;1.50] (*p* < 0.05), and for prosocial problem behavior, it was 1.49 [1.19;1.88] (*p* < 0.001). In addition, the OR for total symptomatic SDQ score was 1.49 [1.22–1.82] (*p* < 0.001). However, the association between hyperactivity symptoms and peer problems was not statistically significant.

#### 3.4.2. Correlations and Differences between Proxy- and Self-Reported Predictors in Adolescent Mental Health Assessments

Spearman’s rho indicated moderate to strong correlations for proxy and self-reported variables: pandemic-related lower family climate at 0.497, pandemic-related adolescent distress at 0.489, days exercising more than 60 min at 0.812, use of digital media for private concerns at 0.753, school concerns at 0.749, and increased use of digital media compared to pre-pandemic levels at 0.485 (all, *p* < 0.001). These results suggest that non-pandemic-related variables have higher correlations between proxy and self-report measures than pandemic-related variables do.

The paired sample Wilcoxon signed-rank test revealed significant differences in perceptions between adolescents and their parents: adolescents rated their family climate more positively (*p* < 0.001) and reported feeling more burdened by the pandemic (*p* < 0.001) than parents (both, *p* < 0.001). Adolescents reported greater use of digital media than their parents did (*p* < 0.001).

No significant differences were observed in reported hours of digital media use in school-related activities. However, for private worries and the number of days spent on physical activities, adolescents’ self-reports were significantly higher than those of their parents (*p* < 0.001).

### 3.5. Parental Mental Health Problems and Parent-Reported Psychosocial Problems

In the 7–10 years age group, the incidence of emotional problems reported by parents without mental health problems was 25.9%, compared with 44.3% for parents with mental health problems (*p* = 0.002). In the adolescent group (11–19 years), these percentages were 27.9% and 60.0%, respectively (*p* < 0.001). For conduct problems, 32.6% of parents without mental health problems reported such symptoms in the younger age group compared with 51.7% of parents with mental health problems (*p* = 0.002), while in the adolescent age group, the figures were 26.9% and 36.0%, respectively (*p* = 0.046). Peer symptoms in younger children were reported by 24.7% of parents without mental health problems and 43.3% of parents with mental health problems (*p* = 0.001). Among the adolescents, the percentages were 28.5% and 40.6%, respectively (*p* = 0.008). Symptoms of hyperactivity were reported by 18.0% of parents without mental health problems in the younger age group, compared to 37.7% of parents with mental health problems (*p* < 0.001). The percentages of adolescents were 13.7% and 27.7%, respectively (*p* < 0.001). For increased problems with prosocial behavior, no significant differences were found in the younger age group between parents without (10.2%) and with (11.7%) mental health problems. However, in the older age group, the percentages were significantly different at 15.5% and 24.0%, respectively (*p* = 0.022). Finally, general symptoms of mental health problems were reported significantly more often by parents with mental health problems in younger children (49.2%) than those without (20.8%, *p* < 0.001), as well as in adolescents (40.2% vs. 21.4%, *p* < 0.001).

### 3.6. Impact of Sports Activity and Private Screen Time Use on Psychosocial Problems

In the younger age group, the number of days spent exercising for more than 60 min per day was negatively associated with several mental health symptoms: emotional problems (Kendall-Tau-b = −0.133, *p* < 0.001), conduct problems (−0.072, *p* < 0.01), social behavioral problems (−0.052, *p* < 0.05), and peer problems (−0.144), and with an overall increased SDQ score (−0.119; both *p* < 0.001), but not with symptoms of hyperactivity. Among adolescents, negative correlations were significant for several symptoms: emotional problems (−0.170, *p* < 0.001), conduct problems (−0.123, *p* < 0.001), hyperactivity (−0.036, *p* < 0.05), peer problems (−0.171), social behavior problems (−0.098), overall higher SDQ score (−0.156), increased anxiety symptoms (−0.141), and depressive symptoms (−0.179; all, *p* < 0.001).

Conversely, increased hours of private screen time in the younger age group were positively associated with symptoms of emotional problems (0.144), conduct problems (0.099, both *p* < 0.001), hyperactivity (0.075, *p* < 0.01), peer problems (0.135, *p* < 0.001), prosocial problems (0.060, *p* < 0.05), and increased SDQ scores (0.140, *p* < 0.001). In adolescents, positive associations were observed for anxiety symptoms (0.157), depression (0.117), emotional problems (0.188), conduct problems (0.173), peer problems (0.138), hyperactivity (0.127), social behavioral problems (0.157), and total symptomatic SDQ scores (0.190; all, *p* < 0.001).

Detailed analyses of gender-specific associations are presented in Figure 2 and Figure 3 for private screen time use and sports activity, respectively.

Figure 2, which shows the relationship between private screen time use and mental health problems, indicates that increased screen time is generally associated with higher levels of emotional problems, conduct problems, hyperactivity, and symptoms of anxiety and depression. This association is significant across all age groups and both genders, with more pronounced effects observed in younger children and adolescents.

Similarly, in Figure 3, which illustrates the associations of sports activity with mental health problems, we observed that increased sports activity generally correlates with lower levels of emotional problems, conduct problems, and symptoms of anxiety and depression. A particularly significant trend is seen for adolescent females, where a spike in frequency at 6 days of sports activity is associated with a notable reduction in mental health problems.

## 4. Discussion

After two cross-sectional surveys were conducted in 2021 and 2022 to assess the psychosocial and mental health impact on children and adolescents in South Tyrol [10,11], a third survey was conducted at the end of the COVID-19 pandemic. The results showed significant mental health challenges among adolescents, with clear differences according to age, gender, and parental mental health. Girls reported higher levels of depressive and anxiety symptoms than boys, with these differences being more pronounced in adolescent groups. Parents reported higher levels of conduct problems and hyperactivity in boys, particularly in younger age groups, whereas girls were more likely to report emotional problems. A key aspect of our findings is the discrepancy between parental and adolescent reports, particularly in perceptions of family climate and pandemic-related stress, suggesting a potential underestimation or misunderstanding of children’s mental states by their parents. In addition, the data highlighted the role of lifestyle factors, with increased physical activity associated with fewer mental health problems, whereas increased screen time was correlated with an increase in several psychosocial problems. These findings highlight the complexity of psychosocial dynamics during the post-pandemic recovery.

The percentage of participants reporting elevated anxiety symptoms was relatively stable over the three years, with 27.2% in 2021, 27.1% in 2022 [10,11], and 27.7% in 2023. A study conducted in Germany using the same measurement tool found that pre-pandemic anxiety levels in children and adolescents were approximately 14.9% [6]. This substantial increase and subsequent persistence of anxiety symptoms throughout the pandemic period underscore the significant and ongoing impact of the pandemic on youth mental health. The finding is concerning because it indicates a persistent level of distress among participants, with implications for long-term mental health. On the other hand, depressive symptoms decreased over the three-year period, particularly among girls. The overall percentage of participants reporting elevated symptoms of depression has decreased from 15.4% in 2021 to 11.8% in 2023. This trend was observed for girls, with the percentage of girls decreasing from 20.2% in 2021 to 13.3% in 2023, whereas the percentage of boys remained stable over the years (10.3% in 2021 to 9.8% in 2023) [10].

Studies have specifically pointed out that depressive symptoms in males increased during the pandemic but have now reverted to pre-pandemic levels [25,26]. This finding challenges the hypothesis that mental health problems will persist or worsen as a long-term effect of the pandemic and suggests possible collective resilience or adaptation among adolescents. However, the persistence of anxiety suggests a more complex picture, with some aspects of mental health recovery while others remain vulnerable. This highlights the need for monitoring and support during future crises, corroborated by the recommendations of Gohari et al. [27].

During the pandemic, school closures and significant family problems were predictive of increased mental health problems in European countries [25,28]. Parent-reported mental health problems in children were strongly associated with pandemic-related variables, namely, an increase in family conflict and inadequate social support, and with the mental health of caregivers [29]. In Italy, depression in children was found to be associated with parental stress [29]. The present findings confirm that mental health symptoms are still associated with pandemic-related factors, such as family climate, child and parental stress, and increased use of digital media. Thus, caution should be exercised by parents and teachers in their daily interactions with the younger generation to detect further mental health problems as potential long-term effects of the pandemic.

Known predictors of mental health problems that were monitored during the pandemic, such as older age and female gender [26,30,31], were confirmed. Single parenthood is a widely known and often discussed factor influencing the mental health of children and adolescents [32], and this factor was confirmed to be significant in our previous studies in 2021 and 2022 [10,11]. In the 2023 study, while low parental education and urban residence were only marginally significant for emotional problems and anxiety symptoms, respectively, no significant effects were found for all other mental health outcomes. However, new associations have also emerged that were not seen in our previous studies conducted in South Tyrol in 2021 and 2022.

First, consistent gender differences across all outcomes confirm the commonly reported differences in boys’ and girls’ experiences [33,34]. In this study, a migration background was defined by whether the respondent or their partner had immigrated to Italy or had foreign nationality at birth. The percentage of students with no migration background in our sample was 84.1%. This figure is comparable to the general population in South Tyrol, where approximately 14.5% of the population has a migration background [35]. This similarity in demographic distribution suggests that the sample is representative of the general population in South Tyrol. Girls consistently reported higher rates of anxiety, depressive symptoms, and emotional problems, while boys showed higher rates of hyperactivity and prosocial problems in both age groups, conduct problems in the younger age group, and peer problems in the older age group. A general increase in mental health problems among adolescents has been described not only during but also before the pandemic, with differences between boys and girls [36,37]. During the pandemic, gender differences in SDQ rates were observed, with a particular increase in symptoms of emotional problems among girls compared to before the pandemic [38]. There were no gender differences in pandemic-related distress in either proxy or self-report measures, but clear gender differences were found in non-pandemic-related leisure behaviors. These findings suggest that in times of crisis, stress is the same for both sexes, but in general, there are differences. Thus, gender-specific interventions should be developed independent of pandemic-related factors. This is an important finding that underscores the relevance of gender-specific research and problem-solving.

Second, migration background emerged as a predictor of depressive symptoms in 2023, a trend not observed during the pandemic [32,36,39]. Migration background was found to be a predictor of symptomatically elevated SDQ, as well as some of its subcategories. This provides a better understanding and allows for the identification of vulnerable subgroups of the population for targeted interventions. In 2023, the survey identified migrant background as a significant predictor of mental health outcomes, a trend that is consistent with findings from an Austrian study in 2022 showing that adolescents with a migrant background face increased mental health challenges, highlighting the need for accessible, culturally and linguistically specific health promotion strategies [37]. This emerging pattern signals a potential shift in sociocultural dynamics affecting mental health in the post-pandemic era. However, when the 2023 data presented here are corrected for multiple testing, this factor is no longer significant for any mental health problem. Thus, we suggest carefully monitoring the mental health development of children and adolescents from migrant backgrounds, knowing that they are a potentially vulnerable group with possible increasing mental health problems. At the same time, one must be aware that there are other significant factors that affect the mental health of adolescents.

Third, the results showed that parental mental health problems were significant predictors of all parent-reported mental health problems in adolescents, while this factor had no significant effect on adolescents’ self-reported mental health outcomes. The present analyses show that proxy reports of parents having a mental health problem themselves nearly double the percentage of children with mental health problems. The only exception was symptoms of problems with prosocial behavior, where no significant difference was found. During the German pandemic, child mental health problems were shown to be associated with parental mental well-being, increased family conflict, and inadequate social support. Parental stress and depression are significantly associated with an increased risk of attention deficit hyperactivity disorder [40,41]. Children with mental illnesses often live with caregivers with mental illnesses [42]. Thus, the association between children’s and parents’ mental health is currently discussed as an important issue [43]. The present results confirm this association. However, while parents clearly reported their own mental health status, the SDQ instrument only reported the symptoms of mental health problems. Parents who themselves experience such problems may be more likely to interpret their children’s behavior as symptomatic. This shared variance due to the single informant effect likely contributes to the significant associations observed. It is important to note that the parent reports of children’s mental health problems were made by the same parents who reported on their own mental health. The results confirm the need to incorporate parental mental health assessments into child mental health intervention practices. By recognizing and addressing the interconnected nature of family mental health, interventions can be more effective and comprehensive, ultimately contributing to better outcomes for both children and their parents.

Fourth, the discrepancy between self-reported and proxy reports highlights the need to consider multiple perspectives when assessing the mental health of children and adolescents. The analysis showed that older adolescents are less likely to complete self-reports and that adolescents with a positive family climate or those who are active in sports are more likely to participate in the assessment process. This suggests a potential selection bias in self-reports, as these factors are associated with fewer symptoms of mental health problems, and parent-reported mental health outcomes were significantly worse among youths who did not complete self-reports. Furthermore, the need for parent reporting is evident, particularly because the most vulnerable subgroups may avoid participating in self-reporting. While parents with mental health problems may over-report their children’s mental health problems, adolescents’ self-reports may under-report them. Therefore, cross-verification of both types of reports is crucial for accurate assessment of adolescents’ mental health. In addition, the significant discrepancies between the parent and child ratings of pandemic-related factors, which do not correlate as strongly as daily behaviors, underscore the importance of careful interpretation of these findings. Several sociodemographic factors contributing to the lack of self-reports from adolescents were identified. Other potential factors may include language barriers, engagement and motivation levels, health conditions, parental influence, school environment, time constraints, or survey fatigue. These factors may systematically influence which adolescents participate in the study and could result in biases. Addressing these issues in future research is suggested to ensure more representative and valid data.

Fifth, the high level of digital media use among children and adolescents is an ongoing concern. Although its use for educational purposes has also decreased since school closures and distance learning in South Tyrol [10,11], screen time for private purposes remains high and has been implicated as a significant predictor of mental health problems [43,44,45,46]. These non-pandemic-related items as well as sports behaviors were clearly associated with adolescents’ mental health, suggesting that the best mental health outcomes are achieved by doing at least 60 min of sports for four or five days a week and using digital media for private concerns one hour or less per day. Using digital media for three hours or more leads to significantly worse mental health outcomes, especially for children up to 10 years of age. Boys aged between 7 and 10 years are highly affected by symptoms of mental health problems when they do not participate in sports. The importance of physical fitness in the development of mental health has recently been confirmed in the context of pandemics [6]. Thus, observations support the recommendation to reduce screen time and exercise for at least three days per week to improve mental health in children and adolescents.

### 4.1. Implications for Public Health and Policy

The findings of this study have important implications for public health strategies and policy development, particularly in the context of global crisis. The persistence of gender disparities highlights the need for gender-specific mental health interventions. The persistence of anxiety symptoms among adolescents highlights the need for sustained mental health support services, suggesting the establishment of accessible, long-term resources, such as counseling services and community-based programs. In addition, the observed reductions in depressive symptoms indicate that resilience can be achieved by promoting supportive community environments and incorporating mental health education into school curricula.

In addition, special attention should be given to vulnerable groups, such as migrant youth and families dealing with parental mental health issues, to ensure that they have access to the necessary support structures and family-wide mental health strategies.

Schools and teachers can operationalize this knowledge by implementing targeted mental health programs and providing additional resources for students with higher needs. In addition, promoting an inclusive environment that recognizes students’ diverse backgrounds and challenges can greatly enhance their support systems and ensure that all students receive the care and attention they need.

Finally, promoting lifestyle changes, such as increased physical activity and reduced use of digital media, could be a straightforward approach to improving overall mental health, particularly among adolescents.

### 4.2. Limitations

The methodology and design of this study provides a dynamic view of the mental health development of children and adolescents. In addition, the nature of data collection, which includes a range of variables from self-reported symptoms to external stressors, allows for the analysis of multiple factors that influence mental well-being.

Despite these strengths, this study has several limitations. The cross-sectional nature of annual data collection is a notable limitation. Although changes can be observed over time, it is not possible to track the progress of the same individual, which limits the ability to make definitive causal inferences. Potential biases were associated with self-reported and parent-reported data. Young participants may have under-reported and parents may have over-reported symptoms. This is particularly relevant in the context of mental health, where stigma or misunderstandings may influence reporting. In addition, although this study included a range of demographic and environmental factors, other unexplored variables may influence mental health outcomes.

Finally, the generalizability of our findings may be limited by the specific demographic or geographical context of the study. Different cultures, healthcare systems, and societal norms may influence the applicability of our findings in other populations. Future research should replicate and extend these findings to diverse settings to provide a more global understanding of post-pandemic mental health.

## 5. Conclusions

This study highlights the dynamic nature of mental well-being in children and adolescents, and the persistence of mental health symptoms. Significant gender differences, with girls consistently reporting higher rates of anxiety, depressive symptoms, and emotional problems and boys reporting symptoms of hyperactivity and problem social behavior, call for gender-sensitive approaches to mental health research, policy, and practice. This study emphasizes the importance of considering a wide range of factors that influence mental health, from individual and demographic characteristics to broader societal and pandemic issues, with a particular focus on the migration background and parental mental health problems. Public health strategies and policies should focus on providing long-term, accessible mental health support, including gender considerations and promoting resilience in children and adolescents. Although not a set of complex solutions, the promotion of prolonged physical activity and the reduction of screen time are basic changes in daily life that can improve the mental health status of the youth. In addition, ongoing research using parallel proxy and self-reported outcomes is needed to further explore the interplay of factors that influence mental health and develop interventions that can address these challenges.

## Figures and Tables

**Figure 1 ijerph-21-00933-f001:**
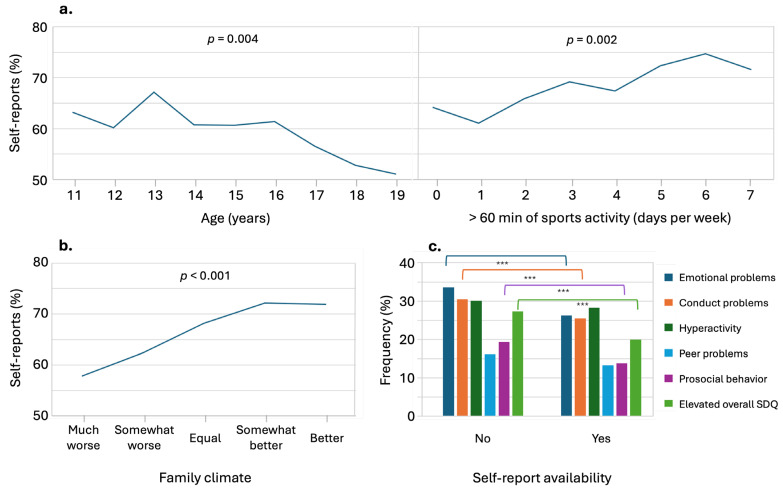
Trends and associations in self-report availability. (**a**) Age (left) and sports activity (right); (**b**) pandemic-related lower family climate; *p* values of Mann–Whitney U test. (**c**) Self-report availability and parent-reported psychosocial symptoms; *p*-values of chi square tests, *** *p* < 0.001.

**Figure 2 ijerph-21-00933-f002:**
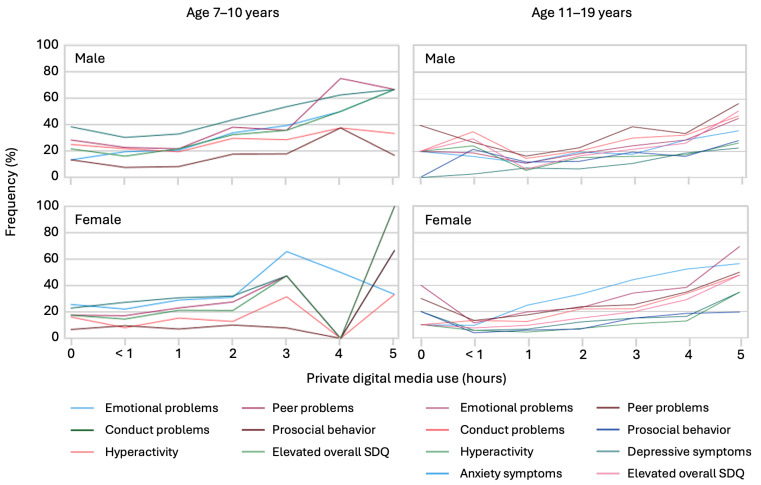
Associations of digital media use for private purposes with percentual prevalence of psychosocial and mental health problems by age and gender. For the younger age group, media use of 4 or 5 h was found only for a few cases, thus percentages assume extreme values.

**Figure 3 ijerph-21-00933-f003:**
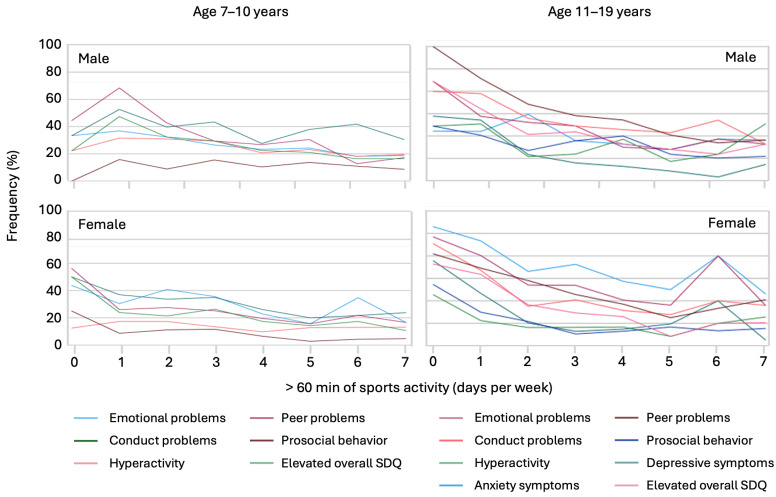
Associations of sports activity with percentual prevalence of psychosocial and mental health problems by age and gender.

**Table 1 ijerph-21-00933-t001:** Sociodemographic characteristics of the child and adolescent mental health sample.

	Children Aged 7–19 Years (Parent Report) *n* = 4525			Children Aged 11–19 Years (Self-Report) ^1^ *n* = 1828		
	M (SD)			M (SD)		
Age of children, years (y)	12.57 (3.55)			14.46 (2.316)		
Age of the parent, years (y)	45.27 (5.28)			47.36 (5.54)		
	%			%		
Age groups	7–10 years 32.9	11–13 years 24.1	14–19 years 43.0	7–10 years -	11–13 years 37.9	14–19 years 62.1
Gender	Male 49.3	Female 50.5	Other 0.2	Male 49.8	Female 49.9	Other 0.1
Gender of the parents	Male 10.1	Female 89.8	Other 0.2	Male 10.1	Female 89.8	Other 0.1
Migration background	No 84.1	Yes 9.5	n.d. 6.4	No 90.6	Yes 8.8	n.d. 0.6
Parental education	Low 22.7	Moderate/high 73.2	n.d. 4.1	Low 25.3	Moderate/high 71.4	n.d. 3.3
Single parenthood	No 89.1	Yes 8.5	n.d. 6.4	No 90	Yes 10	n.d. 0
Residency	Urban 24.5	Rural 75.5		Urban 27.3	Rural 72.7	
Parental mental health problems	Yes 3.8	No 87.2	n.d. 8.9	Yes ^2^ 3.7	No ^2^ 96.2	n.d. ^2^ 0.1
COVID-19 related burden						
Parent: Extreme/reasonably burdened	Yes 35.6	No 55.7	n.d. 8.7	Yes ^2^ 39.3	No ^2^ 60.6	n.d. ^2^ 0.1
Child: extreme/reasonably burdened	Yes 8	No 78.7	n.d. 13.3	Yes 4.7	No 86.8	n.d. 8.5
Lower family climate (much/little)	Yes 18.3	No 71.4	n.d. 10.3	Yes 14.2	No 79.7	n.d. 6.1
More use of digital media (much)	Yes 46.7	No 42.1	n.d. 11.2	Yes 43.9	No 43.1	n.d. 13
Non COVID-19-related						
60+ minutes of sports (3+ days)	Yes 58.9	No 29.8	n.d. 11.3	Yes 59.8	No 28.8	n.d. 11.4
School use of digital media (3+ hours)	Yes 9.1	No 78	n.d. 12.9	Yes 15.3	No 70.5	n.d. 14.2
Private use of digital media (3+ hours)	Yes 60.8	No 27.6	n.d. 11.6	Yes 41.6	No 43.8	n.d. 14.6

^1^ Self-report, if not otherwise indicated; ^2^ Parent report. Unweighted data. Abbreviations: M, mean; SD, standard deviation; n.d., no data.

**Table 2 ijerph-21-00933-t002:** Summary of symptomatic mental health outcomes and predictors by age group.

Variable	Age Group 7–10 Years			Odds Ratio ^1^	Age Group 11–19 Years			Odds Ratio ^1^
	Total (%)	Boys (%)	Girls (%)		Total (%)	Boys (%)	Girls (%)	
Outcomes in Self-reports								
PHQ-2: Symptoms of depression (*n* = 1603)					11.8	10.1	13.3	1.371 [1.008;1.864] *
SCARED: Symptoms of anxiety (*n* = 1565)					27.7	19.3	35.8	2.333 [1.853;2.937] ***
Outcomes in Proxy Reports								
SDQ: Symptoms of emotional problems (*n* = 3715)	26.8	24.5	29.2	n.s.	29.1	25.1	33.1	1.498 [1.256;1.786] ***
SDQ: Symptoms of conduct problems (*n* = 3715)	33.4	36.8	29.9	0.725 [0.572;0.919] **	27.3	28.9	25.8	n.s
SDQ: Symptoms of hyperactivity (*n* = 3715)	19.0	24.0	13.6	0.510 [0.380;0.685] ***	14.2	17.9	10.7	0.543 [0.431;0.685] ***
SDQ: Symptoms of peer problems (*n* = 3715)	25.7	27.6	23.6	n.s.	28.9	30.8	27.1	0.815 [0.685;0.971] *
SDQ: Symptoms of prosocial problems (*n* = 3715)	10.2	11.6	8.8	n.s.	15.5	18.6	12.5	0.624 [0.499;0.780] ***
SDQ: Overall elevated SDQ (*n* = 3715)	22.3	23.7	20.8	n.s.	22.2	22.6	21.7	n.s.
Predictors in Self-reports								
Children’s pandemic-related burden					5.2	4.8	5.5	n.s.
Pandemic-related extended use of digital media					50.4	48.6	52.1	n.s.
Pandemic-related lower family climate					15.0	15.0	15.1	n.s.
Daily use of digital media for school concerns (3+ h)					17.9	15.5	20.2	1.382 [1.067;1.789] *
Daily use of digital media for private concerns (3+ h)					49.1	51.1	47.1	n.s.
More than 60 min of sports a week (3+ days)					67.6	75.1	60.2	0.501 [0.406;0.619] ***
Predictors in Proxy Reports								
Parental pandemic-related burden	37.6	37.4	37.7	n.s.	39.7	40.0	39.4	n.s.
Children’s pandemic-related burden	4.9	4.6	5.2	n.s.	11.4	10.6	12.2	n.s.
Pandemic-related extended use of digital media	40.0	40.7	39.3	n.s.	58.8	58.1	59.5	n.s.
Pandemic-related lower family climate	15.6	16.1	15.0	n.s.	22.7	23.3	22.2	n.s.
Daily use of digital media for school concerns (3+ h)	0.6	0.3	1.0	n.s.	15.1	13.0	17.0	1.371 [1.105;1.699] **
Daily use of digital media for private concerns (3+ h)	7.2	7.1	7.2	n.s.	43.2	44,5	42.0	n.s.
More than 60 min of sports a week (3+ days)	79.2	84.9	73.3	0.491 [0373;0.645] ***	60.1	68.5	52.1	0.499 [0.426;0.584] ***

^1^ Odds ratio refers to the percentage of participants reporting elevated symptoms and female gender. * *p* < 0.05, ** *p* < 0.01, *** *p* < 0.001; n.s., not significant.

**Table 3 ijerph-21-00933-t003:** Summary of non-demographic predictors and unadjusted odds ratios for symptomatic mental health outcomes.

Predictor (OR) ^1^	Symptoms of Anxiety	Depressive Symptoms	Emotional Problems	Conduct Problems	Hyperactivity	Peer Problems	Prosocial Behavior	SDQ Overall
	Self-report	Self-report	Parent report	Parent report	Parent report	Parent report	Parent report	Parent report
Children’s burden (dichotomized)	5.55 [3.418;9.04] ***	9.24 [5.78;14.79]	6.36 [5.02;8.05] ***	3.07 [1.97;4.76] ***	3.07 [2.41;3.91] ***	4.54 [2.898;7.11] ***	2.469 [1.90;3.19] ***	5.87 [4.634;7.43] ***
Extended digital media use (dichotomized)	1.684 [1.34;2.11] ***	2.06 [1.49;2.83] ***	2.16 [1.87;2.50] ***	1.53 [1.33;1.76] ***	1.44 [1.201;1.72] ***	1.61 [1.39;1.86] ***	1.48 [1.23;1.79] ***	2.23 [1.71;2.90] ***
Daily private screen time (hours)	1.21 [1.12:1.30] ***	1.37 [1.22;1.54] ***	1.33 [1.26;1.39] ***	1.18 [1.13;1.24] ***	1.15 [1.09;1.22] ***	1.26 [1.20;1.33]	1.38 [1.30;1.48] ***	1.349 [1.27;1.41] ***
Daily school screen time use (hours)	1.35 [1.21;1.48] ***	1.35 [1.22;1.49] ***	1.14 [1.08;1.20] ***	0.89 [0.85;0.94] ***	0.86 [0.80;0.92] ***	1.10 [1.04;1.16] ***	1.10 [1.03;1.17] **	n.s.
Days of sports activity ≥ 60 min	0.83 [0.78;0.88] ***	0.70 [0.64;0.76] ***	0.81 [0.78;0.84] ***	0.899 [0.86;0.92] ***	n.s.	0.80 [0.77;0.83] ***	0.84 [0.80;0.88] ***	0.81 [0.78;0.85] ***
Lower family climate (dichotomized)	3.93 [2.95;5.23] ***	3.98 [2.84;5.58] ***	5.23 [4.438;6.17] ***	4.01 [3.40;4.72] ***	3.25 [2.69;3.91] ***	2.94 [2.50;3.46] ***	2.52 [2.06;3.07] ***	6.05 [5.08;7.22] ***
	Parent report	Parent report	Parent report	Parent report	Parent report	Parent report	Parent report	Parent report
Parental burden	1.76 [1.41;2.21] ***	1.70 [1.25;2.30] **	2.99 [2.59;3.46] ***	1.90 [1.65;2,19] ***	2.22 [1.86; 2.64] ***	1.89 [1.64;2.18] ***	1.37 [1.14;1.65] **	2.89 [2.46;3.38] ***
Age	1.12 [1.07;1.18] ***	1.26 [1.17;1,34] ***	1.03 [1.01;1.05] **	0.95 [0.93;}.97] ***	0.94 [0.92;0.97] ***	n.s.	1.07 [1.05;1.10] ***	n.s.
Gender	2.33 [1.85;2.94] ***	1.37 [1.01;1.86] *	1.41 [31.22;1.62] ***	0.80 [0.70;0.92] ***	0.53 [0.44;0.63] ***	0.83 [0.72;0.96] *	0.69 [0.57;0.83] ***	n.s.
Single parenthood	1.61 [1.14;2.28] **	n.s.	1.56 [1.23;1.98] ***	n.s.	n.s.	1.71 [2.35;2.17] ***	1.57 [1.17;2.11] **	1.55 [1.20;2.00] **
Migration background	n.s.	1.76 [1.128;2.78] *	1.31 [1.05;1.64] *	n.s.	1.33 [1.01;1,74] *	1.36 [1.09;1.70] **	n.s.	1.31 [1.03;1.68] *
Parental low education	n.s.	n.s.	1.19 [1.01;1.41] *	n.s.	n.s.	n.s.	n.s.	1.22 [1.02;1.47] *
Parental mental health problems	n.s.	n.s.	3.14 [2.28;4.31] ***	1.78 [1.29;2.46] ***	2.58 [1.83;3.63] ***	1.90 [1.38;2.63] ***	1.51 [1.01;2.26] *	2.87 [2.07;3.98] ***
Urban residency	1.33 [1.04;1.70] *	n.s.	n.s.	n.s.	n.s.	n.s.	n.s.	n.s.

^1^ Odds ratios not Bonferroni-corrected. *** *p* < 0.001; ** *p* < 0.01; * *p* < 0.05; n.s., not significant.

## Data Availability

The data presented in this study are available from the corresponding author upon request.

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
