# Peer review of "Parental Mental Health, Gender, and Lifestyle Effects on Post-Pandemic Child and Adolescent Psychosocial Problems: A Cross-Sectional Survey in Northern Italy"

_ijerph, 2024, doi:10.3390/ijerph21070933_

Round 1

Reviewer 1 Report (Previous Reviewer 2)

Comments and Suggestions for Authors

This revised and improved version of the manuscript is suitable for publication. The results are adequately presented and the content is clearly written.

Author Response

We thank the reviewer for this favourable comment.

Reviewer 2 Report (New Reviewer)

Comments and Suggestions for Authors

This study examines parent- and adolescent-reported mental health of Italian children and adolescents near the end of the COVID-19 pandemic.  The paper is generally well-written, and the large sample is a strength.  The manuscript could benefit from addressing the following considerations:

Materials and Methods

·      The paper should describe how parental mental health was measured.

·      Please clarify if the actual use of digital media was reported “per day” (rather than per week, etc.).

Results

·      The source of some of the characteristics in Table 1 are unclear.  My interpretation is that the first set of columns includes parent-reported characteristics, and the second set of columns includes adolescent-reported characteristics.  However, it is stated that some variables (e.g., parent mental health and parent COVID burden) were reported only by parents.  Please clarify the source of reports in the second set of columns.

·      The description of Figure 1 mentions “associations in self-report availability and mental health outcomes”, but 3 of the 4 panels display sociodemographic variables, not mental health.

·      Section 3.5 references Figure 2 as displaying differences in child mental health based on whether or not parents had mental health problems; however, this Figure is not included.

·      The text states that associations with sports activity and screen time are presented in Figures 3 and 4, but these are actually displayed in Figures 3 and 2, respectively.

·      It is very difficult to determine what each line represents in Figures 2 and 3, given the number of outcomes included and the similarity of some of the colors used.  It may be helpful to note which of the outcomes showed unique trends, such as the spike in frequency at 6 days of sports activity for adolescent females.

Discussion

·      The second paragraph of the Discussion notes the stability in anxiety symptoms during the three-year pandemic period.  It would be helpful to also provide an estimate of anxiety levels prior to the pandemic, to support the implied claim in the third paragraph that this persistence of anxiety stems from the pandemic itself.

·      The end of paragraph 5 of the Discussion states that “new associations have also emerged.”  Please clarify it this is referring to associations that were not found in the 2021 and 2022 studies (or something else).

·      The significant associations between parent mental health problems and parent reports of child mental health problems are likely due to the shared variance accounted for by the single informant.  This is somewhat acknowledged at the end of the relevant paragraph, but not clearly.  This interpretation should be noted more clearly in the discussion of this finding, by highlighting that the “proxy” reports of children’s mental health problems were made by their parents, who reported on their own mental health.  Further, clarity would be improved by using the term “parent-report” instead of “proxy-report” throughout the paper.

Author Response

Please se the attachment.

Reviewer 3 Report (New Reviewer)

Comments and Suggestions for Authors

I would like to thank you for your valuable contribution to the field. Your manuscript has the potential to significantly enhance our understanding of Child and Adolescent Psychosocial Problems. I recommend a minor revision to address specific areas that could strengthen your paper, as outlined in the comments below.

Your introduction provides a brief foundation for your study, clarifying the rationale behind the selection of Italy. Ensure that the theoretical framework is coherent and directly linked to your research aims.

In the Materials and methods section, the percentages of students with no migration background in the sample could be further contextualized. How does this figure compare to the general population? This could help in understanding the representativeness of the sample.

In the results section, it would be beneficial to discuss not just the sociodemographic variables but also any patterns or reasons for self-reports of adolescents are lacking, to understand if there were any systematic differences between those who participate and those who lack the study.

The discussion is insightful. The authors could expand on how schools and teachers can operationalize this knowledge to better support students with varying needs.

Finally, I am not sure whether the study's design and the data collection comply with standard ethical guidelines and have proper approval.

Author Response

This manuscript is a resubmission of an earlier submission. The following is a list of the peer review reports and author responses from that submission.

Round 1

Reviewer 1 Report

Comments and Suggestions for Authors

The authors present a study entitled "Long-Term Effects of the COVID-19 on Child and Adolescent Mental Health in Northern Italy".

Abstract

·       At the conclusion of the abstract, there is an omission of the final period.

Introduction

·       This introduction would be strengthened by a complete analysis of the theoretical framework guiding the study. This could include theories on the developmental impact of pandemics on mental health or the role of environmental stressors in psychological outcomes.

·       The introduction provides a brief overview of the impact of the COVID-19 pandemic on child and adolescent mental health. However, it could benefit from a more in-depth analysis of specific studies, particularly those that have longitudinal data, to better establish the context for the current research.

·       The introduction mentions the need for more detailed longitudinal studies in Italy. This could be expanded upon to clarify what specific aspects of child and adolescent mental health have been under-researched in the Italian context.

·       The introduction should present specific hypotheses or research questions that are being tested in the study, derived from the literature review and theoretical framework.

Materials and methods

·       This paragraph is somehow confusing. Please clarify its meaning. “The sample consisted of students aged 7–19 years from public schools in the province. Parents and guardians provided informed consent and participated via proxy questionnaires, while adolescents aged 11–19 were invited to fill out self-report forms.”

·       Authors are requested to provide a more meticulous description of the instruments utilized in the study. This should include the number of items, exemplars of the items, the scale of responses, and indices of reliability for each instrument employed.

·       The use of a repeated cross-sectional design is appropriate for assessing changes over time. It's crucial to check whether the intervals between data collection points are justified and if they adequately capture the trends intended to be studied.

·       The selection of students aged 7–19 from public schools as the sample is a sound choice, but it's important to ensure that the sample is representative of the wider population in terms of demographics like socioeconomic status, ethnicity, and others that may affect mental health outcomes.

·       Within this section, a delineation of the sample should be provided.

Results

·       It is peculiar that the nature of the sample is only revealed within the results section.

·       In actuality, there are three distinct samples, each differing from the others. It remains unclear why identical data would be collected at three separate intervals with different participants, particularly in view of the values reported for the power analysis.

·       The essence of the analyses is decidedly opaque. It is not specified which analysis was conducted to compare, for example, the age differences across the three samples, nor is the purpose of conducting such an analysis.

·       There is a lack of a theoretical rationale underpinning the analyses performed.

·       The reasons why certain analyses are conducted solely for one of the three samples under scrutiny are not elucidated.

·       Furthermore, it should be considered that reporting at least one table to describe the correlations would be appropriate.

·       Additionally, the extensive sample size amplifies the significance of even marginal effects.

Discussion

·       The emergence of new correlations, such as migration background as a predictor in 2023, should be explored further. The discussion should consider why these patterns may have arisen and their broader sociocultural and economic implications.

·       The strong association between children’s HRQoL, psychosomatic complaints, and parental mental health requires a deeper discussion. The implications for family-focused interventions and the importance of considering parental well-being in child mental health assessments should be emphasized.

·       High usage of digital media remains a concern, and its role as a predictor for mental health problems warrants a more nuanced discussion about the challenges and opportunities of digital media in the modern age, especially considering the pandemic’s acceleration of digitalization in many aspects of life.

Comments on the Quality of English Language

The overall quality of the English is sufficient, with only a few minor issues that need checking.

Reviewer 2 Report

Comments and Suggestions for Authors

Review for: ijerph-2758218-peer-review-v1

Some changes and improvements are needed:

Ln. 56-58:  By using validated psychometric tools to measure outcomes, this study aimed to provide critical insights into the mental health trajectory of children and adolescents in northern Italy post-pandemic.

Rephrase (more in English language style) :

By using validated psychometric tools to measure outcomes, this study aimed to provide critical insights into the post-pandemic mental health trajectory of children and adolescents in northern Italy region.

-          the word "trajectory" suggestion to change to some other English phrase more appropriate for this context (for example trend, changes, or just omit this word in this sentence). Through the text change the word "trajectory" with other, more suitable for the context and academic English language writing style.

Decide what word using: sex or gender, carefully check the all text, and correct the text to use the same term throughout the text.

In table 1 is word "gender",  Ln. 102: age and sex; Ln. 129:  sex differences; Ln. 132: gender differences; Ln. 146 and 158, etc…

Table 1.  " burdenedc", " climatec" etc…   the letter c at the end stands for what?

Ln. 158:  title of the Figures: Elevated percentages of proxy…  Omit "elevated" from the title.

Ln. 206: … significant ?? factors.   Positive or negative?

Ln. 221:    3.5. Construction of a Linear regression model

Comments on the Quality of English Language

Suggestions included in previous section.

Reviewer 3 Report

Comments and Suggestions for Authors

Abstract

The abstract must also include data related to the number of participants, gender, age, mean and standard deviation for age, the name of the instruments

Introduction

A description of HRQoL and mention of studies that have analyzed this construct, as well as their results, would be needed, so as to highlight the need for further investigation of these aspects. What does this study bring new? What gaps in the literature does it cover?

The variables to be analyzed should be described. Four hypotheses are presented. Each of these needs a background. Are there other studies showing that returning to normal life after the COVID-19 pandemic led to improved HRQoL? In what context? What could be the additional sociodemographic factors (parental workload, contact with friends, digital media use, children's pandemic burden, plus screen time for private and school-related concerns, physical activity levels)? They must be described. Who might be in the vulnerable groups and why? A brief description of contemporary stressors is required.

Method

Measures

A more detailed description of the instruments is needed and items exaples included (CASMIN and COPSY). How was COPSY modified? Was it validated after the change?

2.2.2 It refers to four instruments. These should be described and items examples presented.

It is unclear whether HRQoL is a stand-alone variable or contains the four dimensions measured separately.

Line 92 – if low HRQol was put in relation with psychosomatic complaints, don't they overlap? Common method variance should be checked.

Line 95 – SPSS must be cited

Results

It should be clarified whether the regression analysis is linear or binomial logistic. If it is binomial logistic it determines the probability that the participants fall into one category or another. What are these categories?

Round 2

Reviewer 1 Report

Comments and Suggestions for Authors

The authors present a study entitled "Long-Term Effects of the COVID-19 on Child and Adolescent Mental Health in Northern Italy".

Introduction:

·       The authors have enumerated several theories that are not directly pertinent to the nature of the study.

·       While the introduction outlines general goals, it is imperative for the authors to present specific hypotheses under examination, derived from a comprehensive literature review and theoretical framework.

Materials and Methods:

·       Authors are urged to provide the response scale and reliability indices for each instrument utilized.

·       The authors state that they have adopted another study's approach to establish intervals between data collection and for comparative analysis. However, in my opinion, there should be a well-defined rationale for these intervals. Mere replication of another approach lacks scientific rigor, especially considering that the primary goal is not to compare results with a previous study.

·       The study, despite encompassing data collection at three distinct points in time, cannot be classified as longitudinal. The explanations provided for this choice do not seem adequate.

·       Questions regarding the Ukrainian situation appear to lack context.

·       The authors should specify the rationale behind conducting a power analysis suggesting the need for 557 participants and subsequently collecting a significantly larger number.

Results:

·       The authors must specify whether both parents can provide responses regarding the same child and clarify if both parents and children are eligible to respond together.

·       There is a lack of a theoretical rationale underpinning the conducted analyses.

·       The reasons why certain analyses are conducted solely for one of the three samples under scrutiny are not elucidated.

·       The table format used to describe correlations is deemed inappropriate.

·       In Paragraph 3.2, it is essential to detail the analyses conducted and report additional parameters beyond p-values.

Discussion:

·       The discussions remain concise and lack a clear rationale.

Comments on the Quality of English Language

The overall quality of the English is sufficient, with only a few minor issues that need checking.

Author Response

Reviewer 1 –  2nd Round

We sincerely appreciate the reviewer's continued efforts and insightful comments aimed at enhancing the quality and integrity of our manuscript. We are grateful for the opportunity to address the points raised and have carefully considered each suggestion. This major revision has led to numerous changes throughout the manuscript. Due to the track-changes mode requested by the editorial office, readability may be somewhat hindered. However, we want to inform you that all new text inserted is exclusively presented within the manuscript itself and not duplicated in this response to the comments. We believe these revisions significantly strengthen our study and provide a clearer, more comprehensive understanding of our findings.

Introduction:

·       The authors have enumerated several theories that are not directly pertinent to the nature of the study.

Thank you for your comment regarding the inclusion of theoretical frameworks in the Introduction. In response to a previous reviewer's request, we incorporated Bronfenbrenner’s Ecological Systems Theory and the Transactional Model of Stress and Coping to provide a theoretical grounding for our study. However, we acknowledge your concern about the direct pertinence of these theories to our research.

To address this, we have trimmed and clarified the text to demonstrate how each theory specifically informs the study design, hypotheses, and interpretation of results. Bronfenbrenner’s theory underpins our approach to examining the environmental influences on children's HRQoL and mental health during the pandemic. Simultaneously, the Transactional Model is presented as the basis for analyzing individual stress responses and their impact on mental health outcomes.

We hope that these revisions make the theoretical foundations of our study more concise and relevant to our research objectives, enhancing the clarity and coherence of the Introduction. Changes to the text are tracked in the revised manuscript.

·       While the introduction outlines general goals, it is imperative for the authors to present specific hypotheses under examination, derived from a comprehensive literature review and theoretical framework.

Thank you for also emphasizing the importance of presenting specific, testable hypotheses derived from the literature and theoretical framework. We recognize that our initial presentation of the study's objectives may have appeared as broad goals rather than precise hypotheses. In response, we have revised this section of the introduction to articulate clear, specific hypotheses that are directly grounded in the literature and informed by our theoretical framework. Changes to the text are reflected in the revised manuscript in Track Changes mode.

Materials and Methods:

·       Authors are urged to provide the response scale and reliability indices for each instrument utilized.

We understand the importance of demonstrating the instruments' reliability to ensure the validity of our findings. In the revised manuscript, we provide the Cronbach's alpha scores obtained for the used survey instruments from our sample in the respective years.

·       The authors state that they have adopted another study's approach to establish intervals between data collection and for comparative analysis. However, in my opinion, there should be a well-defined rationale for these intervals. Mere replication of another approach lacks scientific rigor, especially considering that the primary goal is not to compare results with a previous study.

We recognize the importance of a well-defined rationale for the intervals to ensure the scientific rigor of our study.

The decision to adopt the intervals from the COPSY studies led by Prof. Ravens-Sieberer was made in the context of the unique linguistic and cultural situation of South Tyrol. As a predominantly German-speaking region in Italy, South Tyrol was not included in the national studies conducted in Italy on the psychosocial health of children during the pandemic. In addition, direct participation in the representative studies from Austria or Germany was not possible due to differences in the regional context.

To fill this gap, we collaborated with Prof. Ravens-Sieberer to adapt the methodology of the German national COPSY studies for South Tyrol, ensuring linguistic and cultural appropriateness by providing the questionnaire in both German and Italian. This collaboration allowed a direct comparison of results between South Tyrol and Germany, providing valuable insights into the psychosocial impact of the pandemic on a previously underrepresented minority.

Regarding the intervals between data collection, our decision was influenced by the following factors

(i) Adopting similar intervals to the COPSY studies allowed for direct comparisons between South Tyrol and Germany, providing a richer understanding of the impact of the pandemic in different cultural and linguistic contexts.

(ii) The approximately one-year intervals were chosen to capture the evolving nature of the pandemic and related public health interventions. This timing was considered optimal for observing changes in psychosocial health as the pandemic progressed and public management strategies evolved.

(iii) The recruitment method in South Tyrol, which involved e-mailing invitations to parents or guardians of children attending public schools, differed from the convenience sample in Germany. Despite this difference, the aim was to maintain consistency in the timing of repeated surveys in both regions.

In summary, while our study design has similarities to the COPSY studies, these choices were made with deliberate consideration of the unique context of South Tyrol and the scientific goals of our research. We believe that this approach not only maintains methodological rigor, but also enriches the comparative analysis, providing valuable insights into the psychosocial health of children and adolescents in a region previously unrepresented in national surveys.

In the revised manuscript, we now provide a more comprehensive explanation of these considerations, ensuring a clear understanding of the rationale behind our methodological choices and the unique contributions of our study.

·       The study, despite encompassing data collection at three distinct points in time, cannot be classified as longitudinal. The explanations provided for this choice do not seem adequate.

We acknowledge your concern that the term "longitudinal" may not be the most accurate descriptor for our study, despite the fact that data collection occurs at three different points in time.

We agree that our study is more accurately described as a "repeated cross-sectional" design. Although we collected data at multiple intervals to observe changes over time, each wave of data collection involved a potentially different cohort of participants. This approach does not follow the same individuals over time, which is a hallmark of longitudinal research.

The rationale for choosing this methodology rather than a traditional longitudinal approach was multifaceted: (i) By sampling different cohorts at each time point, we aimed to provide a representative snapshot of the entire population of interest at that point in time, capturing broader trends within the community. (ii) The repeated cross-sectional approach was more feasible given strict data protection regulations in Italy, and allowed us to collect data more quickly and efficiently.

In light of your feedback, we have revised our manuscript to eliminate the term "longitudinal" and to clarify that our study uses a "repeated cross-sectional" design.

·       Questions regarding the Ukrainian situation appear to lack context.

Thank you for pointing out the need for more context regarding questions about the impact of the situation in Ukraine. We acknowledge that without adequate background, the relevance of these questions to our study's focus on mental health and HRQoL may not be immediately apparent.

The inclusion of these items was driven by the recognition that global crises, such as the war in Ukraine, can have profound psychosocial effects on individuals, even those not directly involved in the conflict. As a region bordering Austria and not far from Eastern Europe, the population of South Tyrol may be particularly attuned to developments in Ukraine and the broader geopolitical climate, which could influence mental health and perceptions of well-being. The questions were designed to capture the broader psychosocial environment during the data collection period and to understand its potential impact on the mental health and HRQoL of children and adolescents.

In response to your feedback, we have added additional context to these questions in our manuscript to clarify their relevance and rationale.

·       The authors should specify the rationale behind conducting a power analysis suggesting the need for 557 participants and subsequently collecting a significantly larger number.

First, a power analysis was conducted to determine a minimum sample size of 557 participants to ensure adequate power to detect meaningful effects in our study. This number was based on standard power calculation methods, taking into account expected effect sizes, number of predictors, and desired statistical power.

We were able to enroll a significantly larger number of participants. The inclusion of the initial power analysis in the manuscript text illustrates the power of the survey sample.

Results:

·       The authors must specify whether both parents can provide responses regarding the same child and clarify if both parents and children are eligible to respond together.

In our study, the invitation and information provided to parents and guardians was explicitly designed to encourage participation by both the parent (or guardian) and the child. The survey was structured so that a parent or guardian and the child could provide responses regarding the child's psychosocial health during the pandemic. However, the survey did not specifically accommodate or track responses from both parents for the same child. This approach was chosen to streamline the data collection process and ensure that the survey was manageable for families.

For clarification, additional text was inserted in the Methods section.

·       There is a lack of a theoretical rationale underpinning the conducted analyses.

Our analyses were guided by Bronfenbrenner’s Ecological Systems Theory and the Transactional Model of Stress and Coping. The former helped frame the pandemic's multifaceted impact on children's mental health within various environmental systems. The latter informed our understanding of individual stress responses and their implications for mental health outcomes.

In the revised discussion section, inserted text details how these frameworks shaped our analyses and interpretation.

·       The reasons why certain analyses are conducted solely for one of the three samples under scrutiny are not elucidated.

Our study collected data at three different points in time: 2021, 2022, and 2023. The 2021 and 2022 surveys have been published previously, and these previous analyses provided valuable baseline and follow-up data on the psychosocial health of children and adolescents in South Tyrol during the pandemic. The 2023 survey, presented here for the first time, provides a fresh perspective and allows us to explore more current and relevant aspects of the evolving impact of the pandemic.

As the context of the pandemic and its impact on the psychosocial health of the population have changed over time, some analyses in this study have focused specifically on the 2023 data to capture the most recent reflections of these changes. This approach was also informed by the need to explore new hypotheses and questions that emerged from the current situation and the results of the previous surveys.

In the revised manuscript, we provide a more detailed explanation in the Discussion section of why certain analyses were conducted exclusively for the 2023 sample.

·       The table format used to describe correlations is deemed inappropriate.

Table 2 has been changed to give correlation coefficients and p-values of all variables for each year.

·       In Paragraph 3.2, it is essential to detail the analyses conducted and report additional parameters beyond p-values.

In response to the comment on Paragraph 3.2, the analyses have been detailed extensively, and additional parameters beyond p-values have been reported to provide a thorough understanding of the results.

Tables 3 and 4 have been inserted replacing the supplementary material (all data now presented in the main body of the manuscript.

Correspondingly, the respective text of the results section has been  re-written.

Discussion:

·       The discussions remain concise and lack a clear rationale.

The chapter has been almost entirely rewritten, adopting a structured approach to ensure a comprehensive and clear rationale throughout the discussions.

CJW 7.1.2023